# Preparation of a Consolidation Material of Organosilica-Modified Acrylate Emulsion for Earthen Sites and the Evaluation of Its Effectiveness

**Xin Du [1], Qian Wu [1], Gui Fu [2], Qingwen Ma [2], Guopeng Shen [2] and Hua Li [2,\*]**

[1]    Zhengzhou City Cultural Relic Institute, Zhengzhou 450001, China; duxin1212024@163.com (X.D.); wuqian12120407@163.com (Q.W.)

[2]    School of Chemical Engineering, Zhengzhou University, Zhengzhou 450001, China; fuliuyue0913@163.com (G.F.); mqw2008@zzu.edu.cn (Q.M.); guopen@zzu.edu.cn (G.S.)

\*    Correspondence: lihua@zzu.edu.cn

**Abstract:** In this paper, a new consolidation material for earthen sites with silicone-modified acrylic emulsion was synthesized and applied to the consolidation test of soil samples of the site. The effectiveness was tested through the properties of soil samples on the changes in weight, color, permeability test, air permeability, hydrolysis resistance, water resistance, and salt resistance. The results show that the samples treated with the new material have an outstanding effect on hydrolysis resistance, water resistance, and salt resistance without the change in color and gas permeability. After being soaked in $Na_2SO_4$ and sodium chloride solution for half a month, the reinforced soil sample did not crack, and it could undergo 15 days of water resistance test and five cycles of sodium sulfate resistance.

**Keywords:** earthen sites; consolidation material; consolidation effect; change in property

## 1. Introduction

The development of anti-weathering and consolidation materials for earthen sites is an important aspect in the protection of earthen sites. Most of the earthen sites have been damaged after being buried for a long time, and some of the earthen city walls have weathered and cracked, and there is even a danger of collapse [1–7]. Therefore, the development of a new and effective consolidation material for earthen sites is of great significance to the protection of earthen sites.

As a non-toxic and environmentally friendly polymer material, acrylic resin has the advantages of good film-forming ability, excellent weather resistance, high lightfastness and color retention, strong adhesion, etc., and has outstanding advantages and application prospects in the field of soil site protection. However, traditional acrylic resin emulsion has some disadvantages such as poor hydrophobicity and weak mechanical properties, which affects the practical application value of the material. The Si-O bond contained in the molecular structure of organosilicon compounds has the hydrophobic and heat-resistant properties of inorganic polymer materials, and the Si-O-Si main chain contained in it is relatively soft, not easily decomposed by ozone or ultraviolet light, and has excellent anti-aging properties. In addition, the large size of silicon atoms and the low-cohesion energy density can give the modified polymer excellent anti-fouling and chemical solvent resistance [8–12]. The introduction of silane into acrylate resin can not only combine the advantages of the two, but also improve the water resistance and poor weather resistance of acrylate resin emulsion.

According to the research reports of anti-weathering consolidation materials at home and abroad and the summary of earthen site protection tests [13–16], a new organosilica-modified acrylate emulsion for earthen sites has been developed, and the experimental results show that the consolidation material has a good consolidation effect for the soil site and can meet the various requirements of protection for soil sites.

## 2. Materials and Methods

*2.1. Preparation of Consolidation Materials for Earthen Sites*

(1) Materials

Methyl methacrylate(MMA), Butyl acrylate(BA), β-hydroxyethyl methacrylate (HEMA), Acrylic acid (AA), Glycidyl methacrylate (GMA), Poly(oxy-1,2-ethanediyl),a-sulfo-w-[1-[(4-nonylphenoxy)Methyl]-2-(2-propen-1-yloxy)ethoxy]-,branched,aMMniuM salts (reactive emulsifier DNS-86), Ammonium persulfate(APS), Vinyl triethoxysilane(A-151) are all from Shanghai Maclin Biochemical Technology Co., LTD (Shanghai, China).

Emulsifier aqueous solution A (a mixture of emulsifier and deionized water), Monomer mixture B (a mixture of all monomers), Buffer solution C (a mixture of buffers and initiators), Pre-emulsion D (a uniform mixture of emulsifier aqueous solution A and monomer mixture B) are all from laboratory-made solution.

(2) Preparation of consolidation materials

Add 1.0 g reactive emulsifier DNS-86 and 30–50 mL deionized water to 100 mL round bottom flask and mechanically stir at 40 °C, 350 rpm for 20–40 min to obtain emulsifier aqueous solution A.

Then, 2 g silicone monomer A-151, acrylic monomer (25 g hard monomer MMA, 22 g soft monomer BA, 1 g HEMA and 0.5 g AA and 2 g GMA) were added to 100 mL flask, and the monomer mixture B was obtained by magnetic stirring 5–15 min.

The buffer solution (buffer solution C) of initiator was obtained by stirring 0.25 g initiator APS, 0.25 g NaHCO$_3$, and 30–40 mL deionized water until dissolved.

Preparation of pre-emulsion: the monomer mixture B was added to emulsifier aqueous solution A at a constant speed of 3–4 s/drop rate through a drop funnel, mechanically stirred for 30 min, and then ultrasonic dispersion for 15 min, a stable white pre-emulsion D was prepared.

Preparation of silicone-modified acrylate emulsion: Place 20 g of pre-emulsion D and 10 g of buffer solution C into a 250 mL four-mouth flask equipped with an electric mixer, condenser, thermometer, and drip funnel. When heated to 75–82 °C, blue phase appeared in the reaction, and there was no obvious reflux in the reaction system. At the same time, the remaining 20 g buffer C and the remaining 60 g pre-emulsion D were dropped in, and the drop rate was controlled to be 6–7 s per drop. After about 2.5 h of drip adding, the temperature was raised to 85 °C for 1 h to make the monomer reaction sufficient. After the reaction, the system was quickly cooled to room temperature, and the pH value was adjusted to 7–8 by adding 20% ammonia water. After filtration, the silicone-modified acrylate emulsion with solid content of 35%–38% was obtained. The synthesis of silicone-modified acrylate emulsion is shown in Figure 1.

**Figure 1.** Synthesis of organosilica-modified acrylate emulsion.

*2.2. Preparation of Soil Samples*

(1) Selection of soil sample. The soil samples of a plain reservoir in the Chinese eastern section of the South-to-North Water Transfer Project were selected as the experimental materials, and the soil was screened to remove large particles, then crushed, and the humidity was consistent.

(2) Preparation of standard soil samples. The tool is the compression test mold made by highway geotechnical experiment, which is produced by Beijing Tool Factory. Steel cylinder with inner diameter Φ 50 mm × 20 mm. The compression test mold is used to press the loess into a cylindrical soil sample of Φ 50 mm × 20 mm.

(3) The air-drying of soil samples. The above-mentioned cylindrical soil samples were air-dried in reserve.

*2.3. Consolidation of Soil Sample*

Taking the prepared soil column as the consolidation object, the prepared emulsion was diluted to 10%, 15%, 20%, 25%, 30%, 35% of the solid content, respectively, to strengthen the soil sample, each group of 7 soil column samples.

Consolidation method: The soil sample of the same quality (weight error is not more than 2 g) is soaked in a container of the same specifications with the same weight of reinforcement agent for 20 min and taken out. Then, the sample is placed in air to fully volatilize the solvent and dried at room temperature for one month to test the performance.

*2.4. Determination of Reinforced Soil Samples*

(1) Determination of conversion rate

The conversion rate was measured by gravimetric method. Weigh the emulsion with a certain mass in a test tube and spread it in a dry and clean culture, add a few drops of hydroquinone aqueous solution with a mass concentration of about 5% (to prevent polymerization), and then place it in a drying oven at 90 °C to dry to constant weight before taking it out, and test the 3 samples in parallel to obtain the average value. The formula for calculating monomer conversion is shown in (1):

$$X\% = \frac{M_3 \times (M_2/M_1) - M_4}{M_5} \times 100\% \tag{1}$$

where X% represents the conversion rate, $M_1$ represents the weight of the sample emulsion, $M_2$ represents the mass of the emulsion sample after drying, $M_3$ represents the total mass of the raw material put into the reactor, $M_4$ represents the mass of the non-volatile components put into the reactor, including buffer, initiator and emulsifier, and $M_5$ represents the total mass of the monomer put into the reactor.

(2) Appearance determination

The surface space chromaticity distribution of soil samples before and after reinforcement was measured by the widely used universal portable NR100QC chromaticity meter. Each sample was measured three times, and the color difference value was calculated by the color difference Formula (2):

$$\Delta E^* = \left[(\Delta L^{*2}) + (\Delta a^{*2}) + (\Delta b^{*2})\right]^{1/2} \tag{2}$$

where $\Delta L^*$, $\Delta a^*$, and $\Delta b^*$ are the geometric differences of $L^*$, $a^*$, and $b^*$ before and after specimen reinforcement, respectively, and $\Delta L^*$ represents the black/white difference; $\Delta a^*$ represents the red/green difference, $\Delta b^*$ represents the yellow/blue difference, and $\Delta E^*$ represents the total color difference, which can measure the influence of reinforcement materials on the soil appearance harmony. The smaller the value, the better the appearance compatibility.

(3) Weight determination

A cylindrical simulated sample of the same size ($\Phi$ 50 mm $\times$ 20 mm) was taken, and the dust on the surface of the reshaped sample was gently cleaned with a brush after natural air drying. The blank sample was weighed and recorded as $M_0$. The soil sample was reinforced with emulsion reinforcement agent. After being placed for 15 days under natural conditions, the weight was recorded as $M_1$. The weight difference $\Delta M$ and the change rate $M_d$ of the soil sample before and after reinforcement were calculated, and the calculation formula is shown in (3).

$$\begin{aligned} \Delta M &= M_1 - M_0 \\ M_d &= \frac{\Delta M}{M_0} \end{aligned} \tag{3}$$

(4)  Determination of air permeability

According to the "Test method for water vapor transmission performance of Building Materials and their products" GB/T 17146-2015 determination standard. Determination method: Add the same quality of water (20 g) in the same specification plastic container, cover the reinforced round cake soil ($\Phi$ 50 mm $\times$ 20 mm) on the plastic container with water, and seal the interface with sealant to ensure that water vapor can only be distributed to the outside world through the pores of the sample, resulting in the loss of water quality. Put the soil sample and container into the balance and call their overall mass $G_0$. Put the weighed plastic container and the sample in a cool place with constant temperature and humidity, weigh their mass every 5 days, carry out air permeability measurements for 30 days, respectively, and record the mass as $G_5$. $G_{10}$, $G_{15}$, $G_{20}$, $G_{25}$, $G_{30}$, characterize the air permeability of the reinforced soil sample by calculating the mass of water loss, and measure the air permeability coefficient of the sample. The permeability coefficient of the soil sample is shown in Formula (4).

$$\text{Permeability coefficient} = \frac{\left[\sum \frac{G_i - G_0}{i}\right]}{6} \tag{4}$$

where i = 5, 10, 15, 20, 25, 30, respectively.

(5)  Determination of permeability

The permeability velocity refers to the relationship between the penetration height of the soil and the time. The determination method is that the soil column is upright in the container and the consolidation agent is added to submerge only 2 mm at the bottom of the soil column. When the consolidation agent permeates to the top of the column through capillary force, the ratio of each penetration height and the time is recorded.

(6)  Determination of hydrolysis resistance

Refer to the "Standard for Geotechnical Test Methods" GB/T 50123-2019 determination standard. Determination methods: The soil samples before and after reinforcement were placed into containers filled with water, and the water surface was about 2~3 cm higher than the upper surface of the sample, and the upper part of the container was maintained with air circulation. The water-resistant soaking experiment was carried out at room temperature for 15 days, and the cracking, spalling, and disintegration of the sample were observed and recorded at any time. The immersion height remained unchanged during the experiment.

(7)  Determination of salt tolerance

Refer to the test method of GB/T 50123-2019 "Standard for Geotechnical Test Methods". Determination method: The salt solution used in the experiment was a mixed solution of 5%NaCl + 5% $Na_2SO_4$. The hardened sample was soaked in the salt solution and the height of the solution exceeded the upper surface of the sample by about 3 cm. After being soaked for 12 h, the sample was taken out and dried in a drying oven at 90 °C for 12 h, that is, one cycle was completed, and the change in the sample was observed and recorded.

(8)  Determination of scanning electron microscopy

Phenom LE model field emission scanning electron microscope was used to determine the internal morphology of soil samples before and after reinforcement, and gold spraying treatment was required before sample testing.

## 3. Results and Discussion

### 3.1. FTIR

The acrylic resin latex film before and after modification was determined by infrared spectroscopy to study the structural changes of the copolymer, and the results are shown in Figure 2.

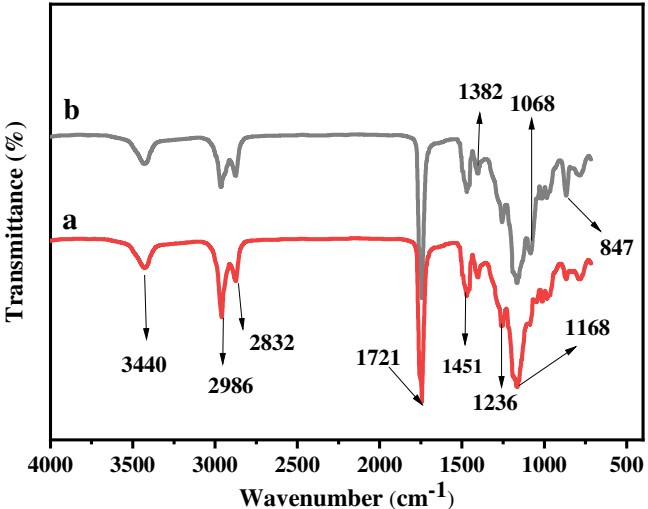

**Figure 2.** Infrared spectra of latex film of acrylic resin and silane-modified acrylic resin: (**a**), Acrylic resin emulsion membrane; (**b**), Silane-modified acrylic resin latex film.

As you can see from Figure 2, a and b of the infrared spectra of the overall trend of the peak are roughly similar. Among them, the absorption peak at 3440 cm$^{-1}$ can be attributed to the stretching vibration of –OH, the absorption peak at 2986 and 2832 cm$^{-1}$ can be attributed to the stretching vibration of –CH in –CH$_3$ and –CH$_2$, and the absorption peak at 1451 and 1382 cm$^{-1}$ can be attributed to the bending vibration of –CH in –CH$_3$ and –CH$_2$. The sharp absorption peak at 1721 cm$^{-1}$ can be attributed to the C=O stretching vibration, and the absorption peak at 1236 cm$^{-1}$ and 1168 cm$^{-1}$ can be attributed to the C–O–C stretching vibration in the ester group. In addition, compared with the infrared map of a, b has a sharp absorption peak at 1068 cm$^{-1}$, and the characteristic strong absorption band of Si–O–C is between 1100 and 1000 cm$^{-1}$, so the presence of Si–O–C can be proved. At the same time, at 847 cm$^{-1}$, the absorption peak strength of b is stronger than that of a, which can prove the existence of Si–C bending vibration, indicating that organosilane participated in the copolymerization reaction [17,18]. Most importantly, there is no obvious expansion vibration peak of the unsaturated double bond C=C between 1500 and 1670 cm$^{-1}$, which indicates that all monomers are basically involved in the polymerization.

### 3.2. Change in Weight

The weight change of soil samples after consolidation is measured according to the difference between the dry weight of soil samples before and after consolidation; Weight change diagram of soil samples before and after consolidation is shown in Figure 3. it is better to change the weight of the soil as little as possible, and the principle of strengthening and protecting soil sites requires a weight change of about 5% [19]. The average amount of consolidation of the six groups of soil samples is 2.12%, 2.39%, 2.68%, 3.48%, 3.71%, and 4.03%, respectively. It shows that the weight of soil increases slightly with the increase in silicone content, but it has little effect on the weight change of soil.

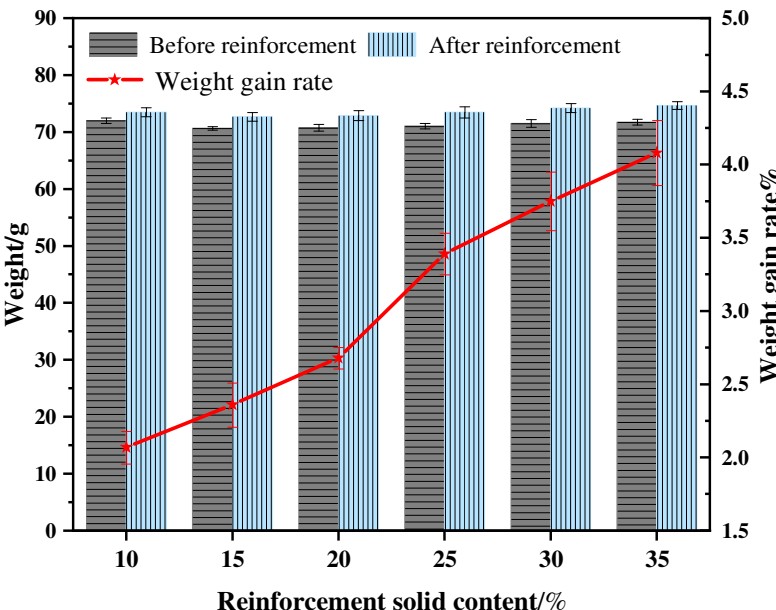

**Figure 3.** Weight change diagram of soil samples before and after consolidation.

### 3.3. Change in Color

An important principle of cultural relics conservation materials is that after the cultural relics are treated with a consolidation material, there is no color change as far as possible [14]. Generally speaking, when the $\Delta E^*$ of the sample is less than five, it can be considered that the color difference is basically unchanged within the visual observation range; that is, the appearance compatibility with the original soil mass is good [4,20].

It can be seen from Figure 4 that with the increase in reinforcement solid content, the $L^*$ value of the sample gradually decreases, while $a^*$ and $b^*$ values show an increasing trend, and the total color difference $\Delta E^*$ gradually increases, which indicates that the appearance of the soil sample gradually becomes black, red, and yellow with the increase in reinforcement concentration, which is consistent with the phenomenon observed by the naked eye (see Figure 5). In addition, when the concentration of the reinforcement agent does not exceed 30%, the total color difference $\Delta E^*$ is less than 5%, and the visual observation shows that the appearance color of the reinforcement soil sample has basically no change compared with the original soil, indicating that the solid content of the reinforcement agent below 30% can meet the requirements of "repairing the old as old" for cultural relics protection. When the solid content reaches 35%, $\Delta E^*$ is greater than five and the surface color of the soil is deepened and black, which does not meet the color difference principle of soil site reinforcement.

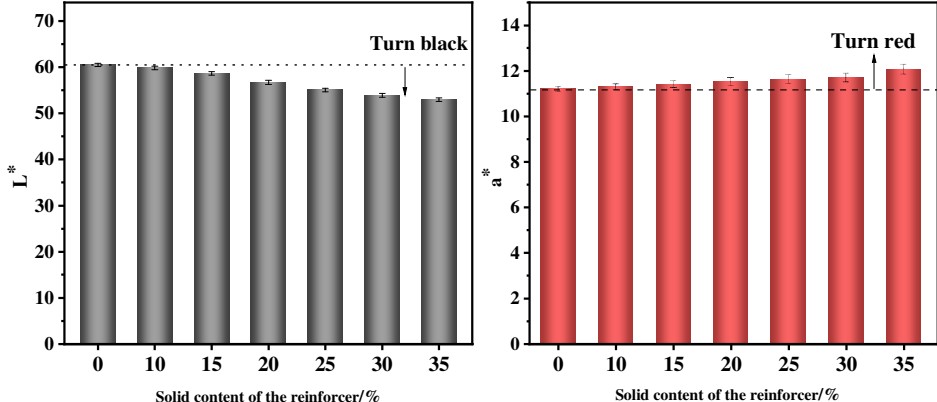

**Figure 4.** *Cont.*

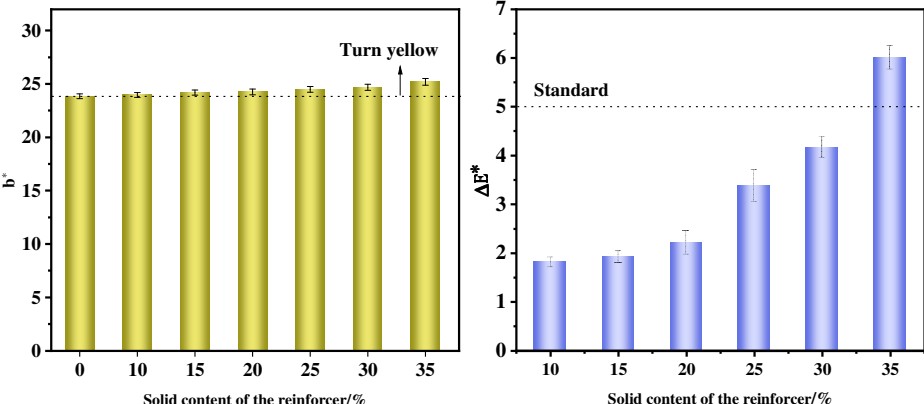

**Figure 4.** The color space value and ΔE* of the sample before and after reinforcement.

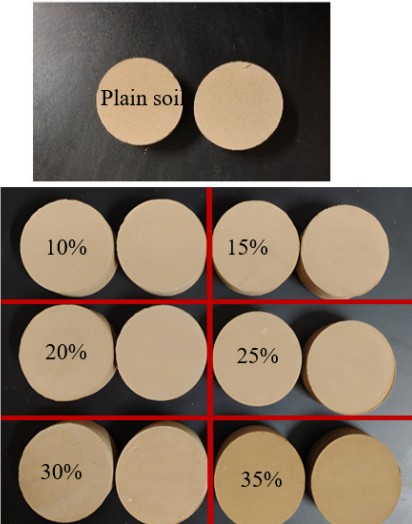

**Figure 5.** The apparent appearance of the specimen before and after reinforcement.

### 3.4. Permeability Test

The permeability velocity is closely related to the porosity of the soil sample and the viscosity of the consolidation agent. The larger the porosity of the soil sample, the smaller the viscosity of the consolidation agent, and the faster the permeability velocity of the consolidation agent permeates to the top of the soil sample through the capillary force. If the permeability velocity is too slow, the consolidation agent has not had enough time to permeate to the top of the soil sample, and the infiltration channel is blocked, thus hindering the further permeability of the consolidation agent, which cannot obtain a good consolidation effect [21]. The permeability velocity of six groups of soil columns is shown in Table 1. It can be seen from Table 1 that it takes a relatively short time for the consolidation agent to permeate to the height of the soil column, 1~5 min, but it takes a relatively long time to fully permeate to the top; in general, the faster the penetration velocity of the consolidation agent to the soil sample, and the better the penetration effect.

It is seen in Table 1 that the initial penetration velocity is relatively fast, but with the increase in time, the penetration velocity decreases obviously because the network structure was gradually formed. By adjusting the ratio of components, a suitable consolidation rate can be adjusted and controlled, and the appropriate penetration velocity and depth can be obtained, which fully shows that the designed system is feasible and effective for the protection of earthen sites.

**Table 1.** Comparison of the permeating velocity.

| Permeating Height, h/mm | Permeating Time, t/s | | | | | |
|---|---|---|---|---|---|---|
| | 10% | 15% | 20% | 25% | 30% | 35% |
| 4 | <1 | <1 | <1 | <1 | <1 | <1 |
| 6 | 2 | 2 | 4 | 5 | 5 | 6 |
| 8 | 5 | 6 | 10 | 10 | 11 | 12 |
| 10 | 8 | 9 | 10 | 12 | 15 | 17 |
| 12 | 13 | 14 | 17 | 23 | 20 | 20 |
| 14 | 14 | 16 | 20 | 25 | 40 | 41 |
| 16 | 20 | 48 | 55 | 58 | 66 | 67 |
| 18 | 31 | 73 | 98 | 101 | 114 | 115 |
| 20 | 63 | 160 | 101 | 223 | 300 | 302 |

*3.5. Air Permeability*

Air permeability refers to the ability of soil samples to circulate water. The consolidation material should not obviously change the air permeability of the cultural relics, ensuring that the large pores in the soil samples are not blocked by the consolidation agent, so that the internal water can communicate in the form of fluid water or water vapor with the outside, and ensure that the cultural relics can breathe freely [7]. The experimental results are shown in Table 2.

**Table 2.** Changes in air permeability of soil samples.

| Consolidation Agent | 5 Days | 10 Days | 15 Days | 20 Days | 25 Days | 30 Days | Mean Permeability Coefficient |
|---|---|---|---|---|---|---|---|
| Blank sample | 5.60 | 9.70 | 16.70 | 21.90 | 27.00 | 28.30 | 1.18 |
| 10% | 4.80 | 9.10 | 14.90 | 17.30 | 23.10 | 26.80 | 1.02 |
| 15% | 4.50 | 9.20 | 13.90 | 18.70 | 21.80 | 25.40 | 0.99 |
| 20% | 4.10 | 8.50 | 12.10 | 16.40 | 19.70 | 24.50 | 0.94 |
| 25% | 4.30 | 7.70 | 11.60 | 14.70 | 21.40 | 23.20 | 0.90 |
| 30% | 3.20 | 7.00 | 11.30 | 15.20 | 18.30 | 20.80 | 0.87 |
| 35% | 3.90 | 7.10 | 9.00 | 12.10 | 16.60 | 18.90 | 0.81 |

It can be seen from Table 2 that the permeability coefficient of the blank sample is the highest; that is, the permeability is the best, indicating that more water vapor passes through the pores of the soil and enters the atmospheric environment, which indirectly reflects that the compact of the internal soil is poor and the inter-particle pores are large. Compared with the blank sample, air permeability decreased after consolidation, but the change was small. It showed that after the soil was treated by the reinforcing agent, the internal pores of the soil block did not obviously hinder the flow of water vapor, and the reinforcing agent basically did not affect the air permeability of the soil. This is because the network structure formed by the reinforcing agent has different mesh sizes, so it does not have a large influence on the air permeability of the soil.

*3.6. Measurement of Hydrolysis Resistance*

Hydrolysis resistance refers to the ability of soil samples to resist water erosion [22]. The test method is that the round cake soil samples before and after consolidation are put into the container with water, and the water is about 2–3 cm higher than the upper surface of the sample, the upper part of the container maintains air circulation, and the water-resistant immersion experiment is carried out in the laboratory environment for 15 days, and the cracking, spalling, and disintegration of the samples are observed at any time, so as to evaluate the water collapse resistance of the soil, and keep the immersion height unchanged in the experimental process. The experimental results are shown in Table 3 and Figure 6.

**Table 3.** Record of water resistance test of soil samples before and after consolidation.

| Serial Number | Experimental Phenomenon |
| --- | --- |
| **Original soil sample** | A large number of air bubbles were produced as soon as the sample was put into water; the soil sample continued to generate bubbles and gradually cracked and disintegrated from the edge at 30 s; the soil sample disintegrated faster and began to accumulate at 60 s, and the clear water became turbid; during the 3rd min, more loose soil accumulated at the bottom of the soil sample, while the structure of the original soil sample basically collapsed; after 5 min, the soil sample immersed in water completely disintegrated and showed a state of scattered sand accumulation. |
| **10%** | Only a few bubbles were generated on the sample surface at 30 s, and after 1 day, the overall structure of the sample was relatively stable, there was no disintegration and collapse, and the surface color of the sample did not change. Three days later, the structure of the soil sample was still integrity and stable, and it was observed that the surface color of the soil deepened and was in a slightly wetting state, and there were bubbles around the soil sample. After 7 days, the structure of the soil sample remained stable, the surface color further deepened, and there were bubbles around the soil sample. After 10 days, the structure of the soil sample remained stable, the surface color further deepened, and there were bubbles around the soil sample. After 15 days, the structure of the soil sample remained stable, and there were bubbles attached around the soil sample. |
| **15%** | It is similar to the A |
| **20%** | There were no bubbles on the surface of the sample at 30 s. After 1 day, the overall structure of the sample was stable, and the surface color was basically unchanged. After 3 days, the structure of the soil sample was still stable, the surface color deepened; it was in a slightly wetting regime, and there were bubbles around it. After 7 days, the structure of the soil sample remained stable, the surface color deepened, and there were bubbles around it. After 10 days, the structure of the soil sample remained stable, the surface color deepened, and there were bubbles around the soil sample. After 15 days, the structure of the soil sample remained stable, and there were bubbles around it in four weeks. |
| **25%** | the same as the C |
| **30%** | the same as the C |
| **35%** | There were trace bubbles on the surface of the sample at 30 s. After 1 day, the overall structure of the sample was stable, the surface color was basically unchanged, and there was a small number of bubbles. After 3 days, the structure of the soil sample was still stable, the surface color deepened; it was in a slightly wetting regime, and there were trace bubbles around it. After 7 days, the structure of the soil sample remained stable, the surface color deepened, there was a very small number of bubbles, and there were bubbles around it in four weeks. After 10 days, the structure of the soil sample remained stable, the surface color deepened, and there were bubbles around the soil sample. After 15 days, the structure of the soil sample remained stable, and there were bubbles around it. |

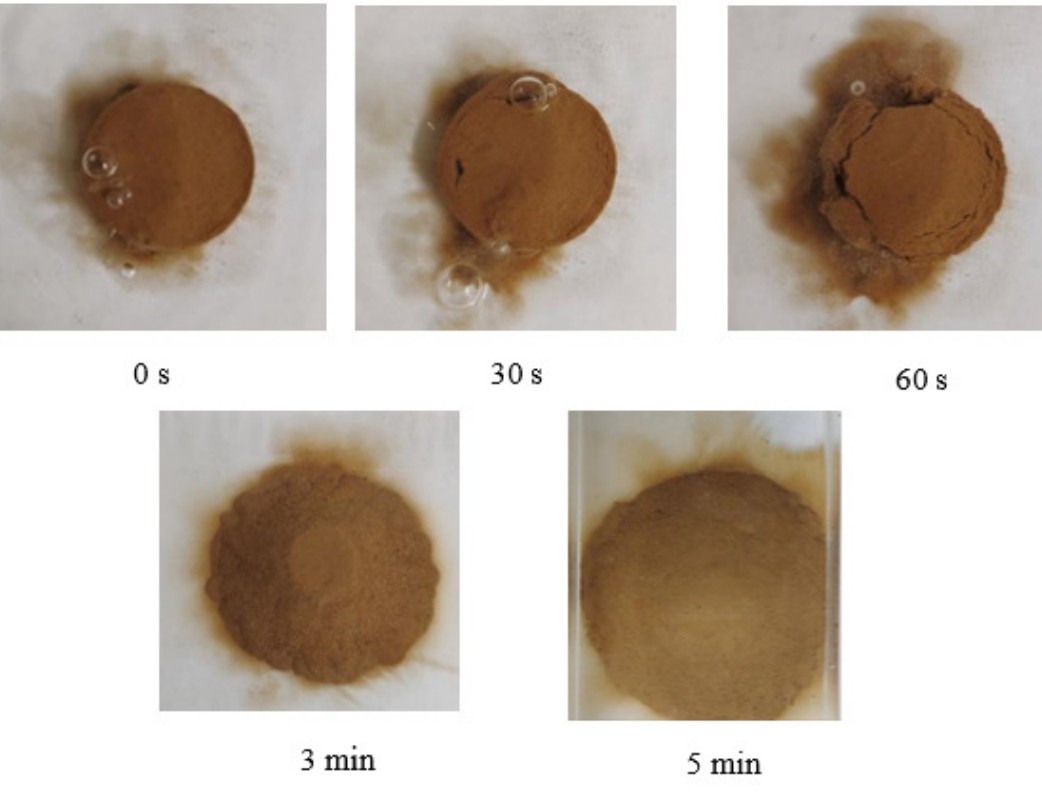

(**a**) Resistance to hydrolysis of unreinforced soil samples.

**Figure 6.** *Cont.*

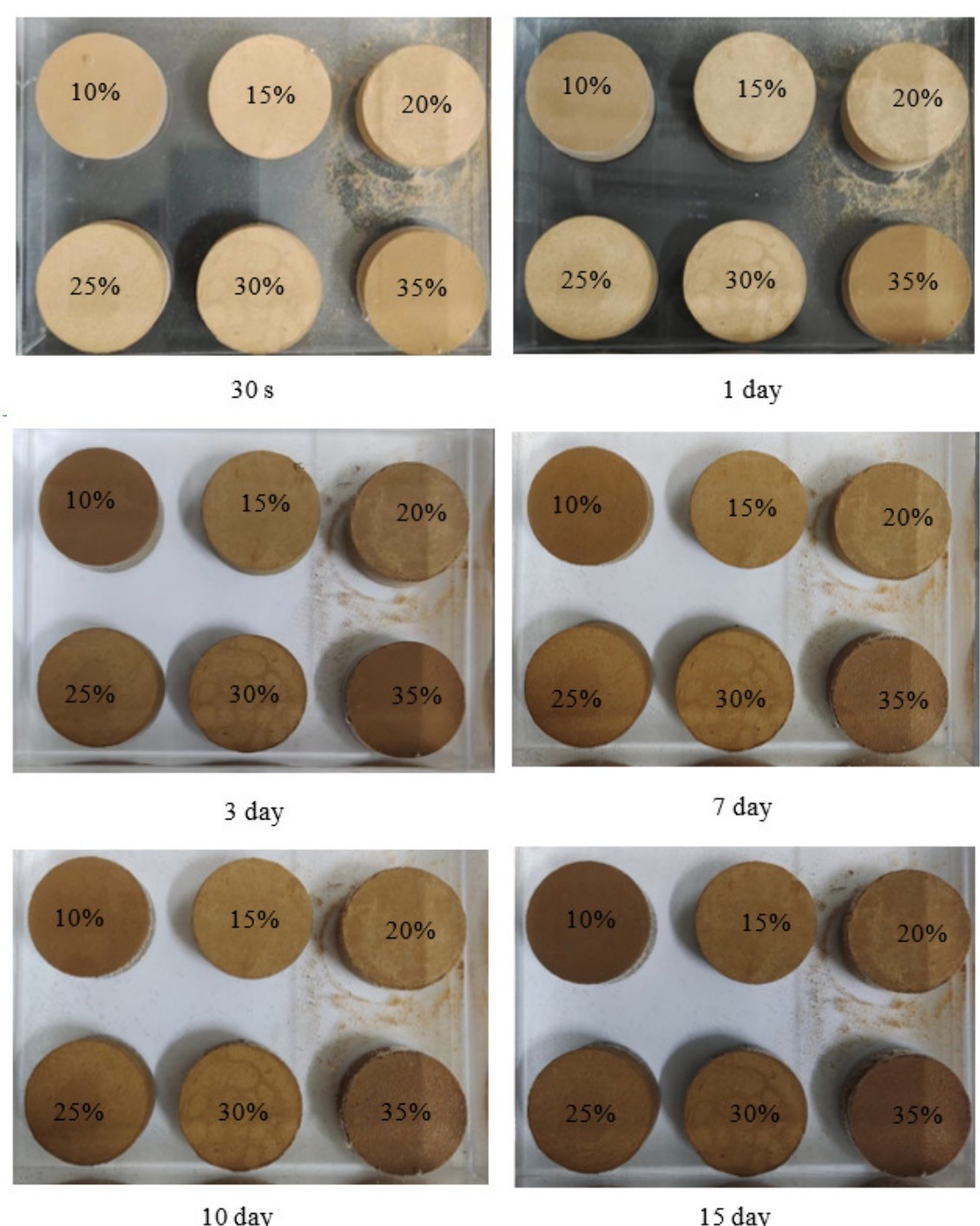

(**b**) The reinforced soil sample is resistant to hydrolysis.

**Figure 6.** Phenomenon of water resistance test for 15 days.

The results of the hydrolysis resistance test showed that the original soil began to absorb water after immersion in water, resulting in a large number of air bubbles, and then began to crack and disintegrate from the edge, and the soil sample completely disintegrated after 5 min. The hydrolysis process of the original soil sample was divided into three stages. The first stage is the absorption stage of a large amount of water: after the sample is immersed in water, it will quickly absorb water and exhaust gas from the pores, resulting in a large number of bubbles. The second stage is the softening stage, the internal pores of the sample are gradually filled and softened by water, so that the softened soil block disintegrates and drops off outward with the bubbles, showing that the edge of the sample

drops off from the outside to the inside. The third stage is the disaggregation stage, where the internal pores of the sample are completely filled with water, resulting in the internal structure of the soil sample becoming very loose, whereby a large number of blocks begin to collapse toward the center, and the hydrolysis rate is accelerated until the soil sample disintegrates completely to an accretion of loose sand. This shows that there were many pores in the original internal soil sample, inter-particles were loose, the granular structure was seriously damaged when mixed with water, the pore structure of the soil was also seriously damaged, and water resistance was poor.

After coating with a silicone-acrylic emulsion with different solid contents, there were only a small number of bubbles, and no disintegration or drop-off phenomena were produced in the soil samples, and the overall structure of the soil sample was still stable after 15 days of water soaking. Its microstructure did not obviously change, and the water resistance of soil samples was higher, which indicates that the synthesized silicone-acrylic emulsion could enhance the hydrolysis resistance of soil sites. This may be due to the fact that after the volatilization of the solvent, the reinforcing agent permeates into the soil sample and forms the thin film, and some polymers form a cross-linked structure after solidification; in addition, the Si atoms in the consolidation agent have a certain hydrophobic effect, which can migrate to the surface of the membrane, resist the invasion of water molecules, and increase the hydrolytic resistance of the soil surface [23].

*3.7. Measurement of Salt Resistance*

Under the action of long-term wind erosion and rain erosion, the salt in the soil site is taken away or flows to the bottom, and under the action of capillary water, the salt content increases, the soluble salt content of the soil increases, and the corrosiveness increases. Due to the rise of capillary water, erosion occurs at the bottom of the soil, so the salt resistance is also an important index to evaluate the solidification effect [24]. The experimental results are shown in Figure 7.

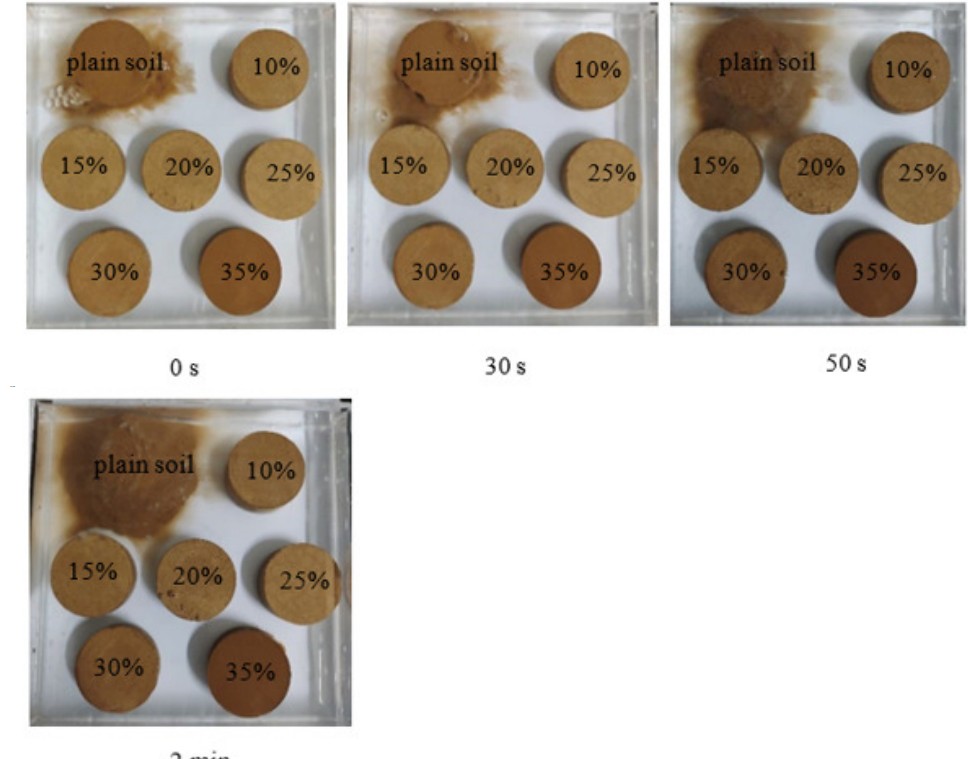

**Figure 7.** Disintegration phenomenon of salt resistance test of soil samples treated by different solid content consolidation agents.

In the salt resistance test, after soaking the original soil sample in saturated $Na_2SO_4$ and NaCl solution, a large number of air bubbles appeared at first, then began to collapse from the edge, and then completely disintegrated at 90 s. The salt crystals migrated to the internal pores of the soil sample, resulting in the destruction of the internal structure and salt resistance became poor. The main reason for this is due to the reaction of $Na_2SO_4$ and water. When the water in the soil increases, $Na_2SO_4$ can be crystallized into $Na_2SO_4 \cdot 10H_2O$, which is about three times larger than that of anhydrous $Na_2SO_4$, which causes pressure on the internal pores of the soil and destroys the internal structure of the soil. When the soil is in a dry environment, $Na_2SO_4 \cdot 10H_2O$ will lose water and convert into anhydrous $Na_2SO_4$, and the volume of salt becomes smaller. In the process of interconversion, the internal structure of the soil is destroyed under the action of the salt swelling force, which is the fundamental reason for the decrease in sample strength after the salt resistance test; the salt resistance of earthen sites can be increased greatly after adding consolidation materials.

### 3.8. Scanning Electron Microscopy

It can be seen from Figure 8 that the structure of the raw soil is loose and has more pores, while the structure of the treated soil is more compact and the porosity is decreased. From the comparison of the microstructures, it can be seen that there are some changes in the internal structure of the treated soil; they mainly manifest in the phenomenon that soil particle packing is more dense, which attributes good compatibility of the organosilica to soil; the dense network structure can effectively fill the pores in the soil samples and play a certain bridging role, which increases the binding force of the soil particles and reinforces the soil.

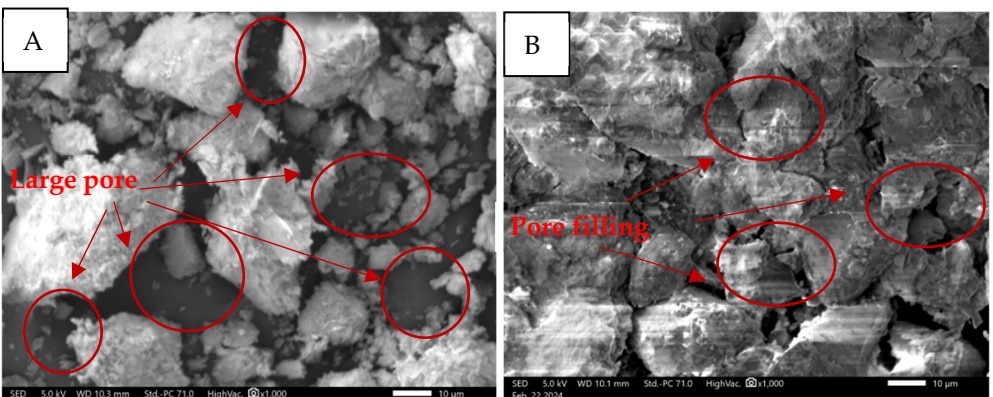

**Figure 8.** SEM of the soil samples: (**A**), raw soil; (**B**), treated soil.

### 4. Conclusions

The organosilica-modified acrylic clay consolidation material is prepared and applied to the consolidation test of soil samples of earthen ruins. The results show that the synthesized consolidation material has a good consolidation effect on soil samples. After consolidation, the weight, color, and air permeability of the soil have no obvious changes, which basically does not affect the appearance and original properties of the soil, and can obviously improve the acid and salt resistance of the soil, as well as the hydrolysis resistance and stability of the soil. Some reinforced soil samples will not crack after 10 hydrolysis resistance cycles and 5 salt resistance cycles. SEM analysis shows that the particle accumulation of the coated soil is dense and the porosity is decreased, indicating that the coated material effectively fills the pores in the soil and plays a supporting role; the network structure formed by the reinforcing agent has different sizes of mesh, which will not have a great impact on the air permeability of the soil, and there are still many micropores in the reinforced soil, so the air permeability of the reinforced soil changes a little bit. In addition, by comprehensive comparison, the reinforcement effect of 25% consolidation percentage

is better. By introducing different contents of organosilicon, the consolidation effect was obviously improved with the increase in silicone content.

**Author Contributions:** Conceptualization, H.L. and X.D.; methodology, H.L. and G.F.; validation, Q.W., G.S. and Q.M.; formal analysis, G.F. and X.D.; investigation, H.L. and G.F.; resources, H.L. and G.F.; data curation, X.D. and G.F.; literature retrieval and chart making, G.F.; writing—original draft preparation, H.L. and X.D, revision of manuscripts, H.L.; writing—review and editing H.L and X.D.; supervision, H.L. All authors have read and agreed to the published version of the manuscript.

**Funding:** This research received no external funding.

**Institutional Review Board Statement:** Not applicable.

**Informed Consent Statement:** Not applicable.

**Data Availability Statement:** Data are contained within the article.

**Conflicts of Interest:** The authors declare no conflict of interest.

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
