# Peer review of "Preparation of a Consolidation Material of Organosilica-Modified Acrylate Emulsion for Earthen Sites and the Evaluation of Its Effectiveness"

_coatings, doi:10.3390/coatings14050587_

Round 1

Reviewer 1 Report

Comments and Suggestions for Authors

Line 21-30: The introduction is to simple. Current solutions and their disadvantages are not stated. Why this organosilica-modified acrylate emulsion offers better solution should be included.

Line 33-37: Poor editing and formatting

Line 41: What is solution A?

General comment on materials and methods section: The authors need to clarify all the solutions and mixtures A-D. Give introduction first. Some chemicals stated are not mentioned their source of origin.

Figure 1: Need to mention the temperature, time, pH and other necessary parameters important to polymerization.

Line 87: How little this should be?

Line 94-98: An explanation of what does the color indicates need to be elaborated well. If this is not explained with proper examples and examples, this is worthless.

Line 103-114: No citations/references to backup these statements.

Line 201-208: This portion should be in the materials and methods section.

Line 226-233: How do the authors  define compactness and looseness here? No measurements are stated and quantified.

Solid conclusion on which consolidation percentage gives the best results are not mentioned at all.

Comments on the Quality of English Language

The language quality is good.

Reviewer 2 Report

Comments and Suggestions for Authors

The submitted article can be recommended for publication after minor revision.

For all details see the attached file.

Comments on the Quality of English Language

The English style is acceptable. Minor corrections are recommended  in the attached file, too.

Reviewer 3 Report

Comments and Suggestions for Authors

This paper deals with the preparation of a copolymer emulsion by emulsion copolymerization of six different monomers, and using the resulting emulsion for consolidation of earthen samples by soaking these in the emulsion. This manuscript is recommended to be accepted for publication after some revision on the basis of comments below.

COMMENTS

1.

In line 35, emulsifier is mentioned as DNS-86. The full chemical name and chemical structure of this emulsifier should be provided in this manuscript.

2.

What is APS (line 46). It is not mentioned in the Materials section on page 1.

3.

What the authors mean on “blue phase” in line 53?

4.

The authors do provide exact volumes for the water used to make the solutions. They give 30-50 mL for the emulsifier solution and 30-40 mL for the initiator+NaHCO3 solution. Furthermore, 1/3-1/2 and 1/2-2/3 of the solutions are given for carrying out the polymerization. In order to reproduce the results of the authors, exact amounts should be provided.

Reproducibility is the most important requirement of scientific and technical/technological publications !

5.

It is not written why the authors used dropping of the second portion to the first one at 75-82 C. Emulsion polymerizations can be carried out without such a two stage process.

6.

The time required for the addition of the second portion of the components by dropping during the applied process should be provided.

7.

The authors write that keeping the temperature at 85 C for one hour at the end of the process “make the monomer fully react”. How the authors determine the conversion of the monomers? There is nothing written about the conversion of the monomers.

8.

Did the authors determine the composition of the prepared copolymer? The data on the copolymer composition should be provided.

9.

It is important to provide explanation why the authors selected the listed monomers for making an emulsion of the copolymer formed from these monomers. This should be provided.

10.

The authors should also explain why the following amounts of the comonomers were used (in grams):

MMMA         25

BA                22

AA                0.5

HEMA 1

GMA            2

A-151 2

11.

In subsection 2.3, the authors do not provide the experimental details correctly for the consolidation method. No any data is given on the weight of the soil samples and the amount (or volume) of the soaking emulsion with different solid contents. A table should be provided on the exact amounts used for the soaking experiments.

12.

It is absolutely unclear what the number mean on the y-axis on the right.

13.

In Table 2, it is absolutely unclear what the number means under the 5d to 30d, on the one hand. On the other hand, the authors do not provide any description on the determination process of the permeability coefficients. Third, the numbers with three decimal digits precisity are unacceptable. What are the errors of these data?

14.

In Table 3, what the authors mean on “hydrophobic bubbles”? This is nonsense in an aqueous system.

15.

In lines 192-193, the authors write that “the Si atom in the consolidation agent has a certain hydrophobic effect” (correctly has and not have). Considering that the monomer mixture for preparing the copolymer contains altogether 93% hydrophobic comonomers of MMA, BA and GMA (see Experimental in the manuscript and Comment 10), these definitely have much higher “hydrophobic effect” than the vinyl silane comonomer (3.8%) with three ethoxy groups, which undergo hydrolysis anyway under the applied conditions. The authors should explain why they think that the vinyl silane with minor amount should have exclusively “hydrophobic effect”.

16.

In the Experimental, the details of the scanning electron microscopic measurements are not described. This should be added.

17.

The title of subsection 3.7 correctly: Scanning Electron Microscopy

18.

The format of the list of the references on page 11 is a mess. This has to be corrected according to the requirements of the journal. See the “References” subsection for the rules of manuscript preparation at the following website:

https://www.mdpi.com/journal/coatings/instructions#preparation

All the references are in wrong format. For instance, References 8-12 in the manuscript even do not have the name of the authors. In contrast, some other references (Refs. 16-18) have full names.

Author Response

plaese see the attachment

Round 2

Reviewer 1 Report

Comments and Suggestions for Authors

The reviewer had read the new manuscript and found this version is more sound and better than the previous. The scientific merit of the paper is fine and may attract attention of possible audience of the field.